
# Asymmetric dark matter: residual annihilations and self-interactions

Iason Baldes[1][*], Marco Cirelli[2], Paolo Panci[3], Kalliopi Petraki[2,4], Filippo Sala[1] and Marco Taoso[5,6]

**1** DESY, Notkestraße 85, D-22607 Hamburg, Germany
**2** Laboratoire de Physique Théorique et Hautes Energies (LPTHE),
UMR 7589 CNRS & Sorbonne Université, 4 Place Jussieu, F-75252, Paris, France
**3** CERN Theoretical Physics Department, CERN, Case C01600, CH-1211 Genève, Switzerland
**4** Nikhef, Science Park 105, 1098 XG Amsterdam, The Netherlands
**5** Instituto de Fisica Teórica (IFT) UAM/CSIC, calle Nicolás Cabrera 13-15,
28049 Cantoblanco, Madrid, Spain
**6** INFN, Sez. di Torino, via P. Giuria, 1, I-10125 Torino, Italy

★ iason.baldes@desy.de

## Abstract

Dark matter (DM) coupled to light mediators has been invoked to resolve the putative discrepancies between collisionless cold DM and galactic structure observations. However, $\gamma$-ray searches and the CMB strongly constrain such scenarios. To ease the tension, we consider asymmetric DM. We show that, contrary to the common lore, detectable annihilations occur even for large asymmetries, and derive bounds from the CMB, $\gamma$-ray, neutrino and antiproton searches. We then identify the viable space for self-interacting DM. Direct detection does not exclude this scenario, but provides a way to test it.

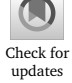

# 1   Introduction

There is a plethora of proposals explaining dark matter (DM) as an exotic, beyond-the-Standard-Model particle. Cold DM can explain the galactic rotation curves [1, 2], velocities of galaxies in clusters [3], lensing observations [4, 5], and the combination of observations ($X$-ray, lensing, visible) in galaxy cluster collisions [6]. Further evidence comes courtesy of the acoustic peaks in the CMB. These measure the amounts of baryonic matter, i.e. coupled to the radiation bath, and matter decoupled from the radiation bath in the early Universe [7]. On the other hand, searches for DM-induced nuclear recoils at direct detection experiments [8–13], production of DM at colliders [14–16] and indirect detection of DM through observation of its annihilation products in the universe around us [17, 18] — apart from some long-standing anomalies [19–26] — have so far turned out negative.

*Self Interacting Dark Matter.* Another probe of the DM properties is the structure DM forms. Observations of the large-scale structure are in good agreement with theoretical expectations of the clustering of collisionless cold DM [27]. However, discrepancies arise on small, sub-galactic, scales. These include the core-cusp [28–30], too-big-to-fail [31, 32], missing satellites [33, 34], and diversity problems [35]. It is possible that baryonic feedback effects, which are difficult to accurately model, may alleviate these issues [36–38]. Alternatively, the resolution may lie within the physics of the dark sector. If the DM particles self-scatter significantly inside halos, they redistribute energy and momentum, thereby affecting the galactic structure [39, 40]. Further observational and theoretical work must be done to determine whether baryonic effects or self-interacting DM (SIDM) is the solution to the small scale structure problems. If the solution is indeed SIDM, this would provide a powerful tool in determining the underlying particle physics model.

A minimal realization of SIDM is fermionic DM coupled to a dark $U(1)_D$ gauge force. The dark vector boson $V$ may acquire a mass via either the Stückelberg or the Higgs mechanisms. A small but finite mediator mass allows sizable DM self-scattering while ensuring that the interaction is not dissipative [41–43].

*Indirect detection constraints on SIDM.* Such models, when the DM abundance is set by thermal freeze-out, are however highly constrained by a combination of astrophysical and cosmological observations. This is because, in the simplest scenario, the dark mediators produced in the DM annihilations, decay into SM particles (photons, electrons/positrons, protons/antiprotons, neutrinos) through a kinetic mixing with the hypercharge gauge boson.[1] Final states with SM particles imply strong constraints from DM indirect detection and, most

---

[1] Decays to the SM, or to another form of radiation, are necessary to avoid that the dark mediators dominate the energy density of the Universe when they become non-relativistic [44].

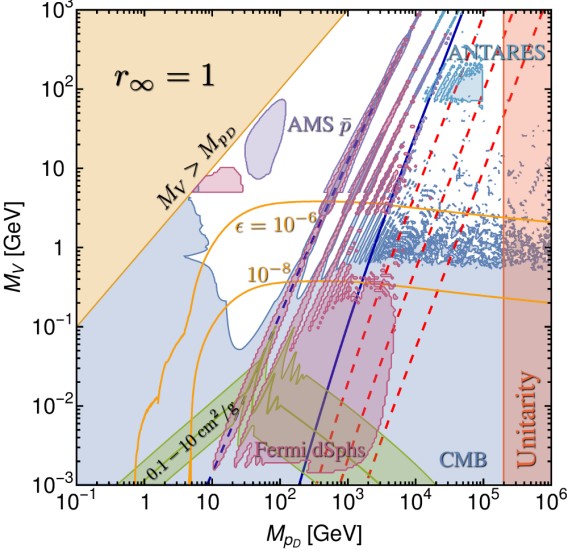

Figure 1: Constrained regions (shaded areas in blue, pink, purple and cyan) in the parameter space of the DM mass $M_{p_D}$ and mediator mass $M_V$, together with areas of sizable self-interactions (shaded green), for the symmetric DM case, i.e. $r_\infty = 1$, and including the effect of the dark electrons. The areas of sizable self-interactions are ruled out by the CMB and FERMI dSphs constraints. This panel is to be contrasted to the asymmetric cases, $r_\infty < 1$, illustrated in Fig. 4. The yellow contours show the direct detection constraints for the indicated values of $\epsilon$. The blue dashed line shows the location of the first parametric resonance due to the Sommerfeld effect. Formation of $\bar{p}_D p_D$ bound states is possible to the right of the blue solid line. Formation of $p_D e_D$ bound states is possible to the right of the red dashed lines for, from left to right, $m_{e_D} = M_{p_D}/10, 100, 1000$. The kinematically inaccessible part of the parameter space on the top left corner, shaded here for $M_V > M_{p_D}$, extends to encompass the region $M_V > m_{e_D}$ in the presence of dark electrons.

importantly, from energy injection in the CMB. Indeed, Refs. [44,45] recently showed that the parameter space with sizable self interactions is ruled out, as reproduced in Fig. 1.[2]

*Enter a dark asymmetry*. In this paper, we entertain a slightly more complex possibility. We introduce a particle-antiparticle asymmetry in the dark sector, in analogy to and partly motivated by the ordinary baryon asymmetry [46,47]. The density of asymmetric DM (ADM) is determined, at least partly, by the excess of DM particles over antiparticles, $Y_D \equiv Y_+ - Y_-$ where $Y_+$ ($Y_-$) is the DM (anti)particle-to-entropy density ratio.

The presence of an asymmetry suppresses the indirect detection signals [48–50], simply because there are fewer antiparticles to annihilate with the DM particles. Hence, large ranges of the DM and mediator masses that are ruled out in the symmetric DM scenario, are viable in the presence of an asymmetry. However, an accurate computation reveals that even highly asymmetric dark matter with long-range interactions can produce significant indirect detection signals, due to the residual DM annihilations being enhanced by the Sommerfeld effect. This is a non-trivial result, first pointed out in [51], which deserves further scrutiny.

The purpose of this paper is therefore twofold: (i) to compute indirect detection constraints on ADM with long-range interactions; (ii) to identify the parameter space where all constraints

---

[2] The constraints in Fig. 1 are updated with respect to those of [44,45], to include the effect of an additional lighter species of dark fermions, which we shall refer to as *dark electrons*, see below.

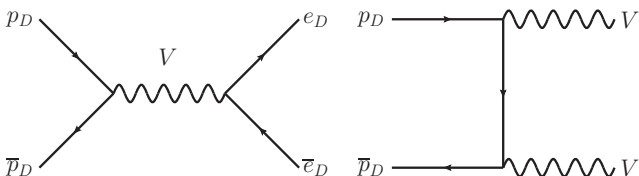

Figure 2: Dark matter annihilation processes. Due to the light mediator the annihilations are Sommerfeld enhanced. Unstable $p_D - \bar{p}_D$ bound states can also form radiatively and decay into mediators, thus depleting the DM density.

are evaded, while DM self-interactions are sizable.

The rest of this paper is organized as follows. In Sec. 2, we specify the details of the model. In Sec. 3 we briefly discuss the processes which set the relic density. In Sec. 4 we compute the DM self interactions, before deriving the indirect detection constraints in Sec. 5. Direct detection is discussed in Sec. 6 (providing updated constraints to [52,53]). Important caveats regarding the formation of dark atomic bound states and the reannihilation effect are explained in Sec. 7. We then briefly comment on our findings and conclude.

## 2 The Model

We consider a dark QED-like model, in which DM consists of Dirac Fermions that couple to a dark gauge $U(1)_D$ force. If DM also carries a particle-antiparticle asymmetry, then gauge invariance implies that there must be at least two dark species with compensating asymmetries, such that the total gauge charge of the universe vanishes. While this is evident in the case of an unbroken $U(1)_D$, it remains inevitable under reasonable assumptions if $U(1)_D$ is mildly broken. Importantly, this encompasses the parameter space in which the dark vector boson is sufficiently light to mediate sizable self-interactions that can affect the galactic structure [54].[3]

We shall thus introduce two dark species, the dark protons $p_D$, and the dark electrons $e_D$, that couple with opposite charges to the dark vector mediator $V$. For our purposes, both $p_D$ and $e_D$ are elementary point-like particles with masses $M_{p_D}$ and $m_{e_D}$, and $M_{p_D} \gg m_{e_D}$. The Lagrangian is

$$\mathcal{L} = \frac{1}{2} M_V V_\mu V^\mu - \frac{1}{4} F_{D\mu\nu} F_D^{\mu\nu} - \frac{\epsilon}{2c_w} F_{D\mu\nu} F_Y^{\mu\nu} + \bar{p}_D(i\slashed{D} - M_{p_D})p_D + \bar{e}_D(i\slashed{D} - m_{e_D})e_D, \quad (1)$$

where $D^\mu = \partial^\mu \pm i g_D V^\mu$ is the covariant derivative for $p_D$ and $e_D$. The field strength tensor is $F_D^{\mu\nu} = \partial^\mu V^\nu - \partial^\nu V^\mu$, and $\alpha_D \equiv g_D^2/(4\pi)$ is the dark fine-structure constant. In the case we consider here, there is a conserved dark baryon number associated with $p_D$ and a conserved dark lepton number associated with $e_D$. A linear combination of dark baryon and dark lepton number needs to be broken at high energies for the generation of the asymmetry, and the inclusion of the dark electrons means this can be achieved in a gauge invariant way, in analogy with higher dimension baryon violating operators added to the SM. For concreteness we assume here that the Stückelberg mechanism generates $M_V$. The introduction of a dark Higgs is not expected to change our conclusions, beyond minor numerical differences. It is important to note though, that the assumption of asymmetric DM implies that the Higgs mechanism does not generate a Majorana mass for at least one of the dark species, here the dark protons. This only restricts the $U(1)_D$ charge of the dark Higgs. For more complete models, see e.g. [55–58].

---

[3]If $U(1)_D$ breaks via the Higgs mechanism at a temperature above the dark asymmetry generation, then an asymmetric population of dark electrons need not be present today.

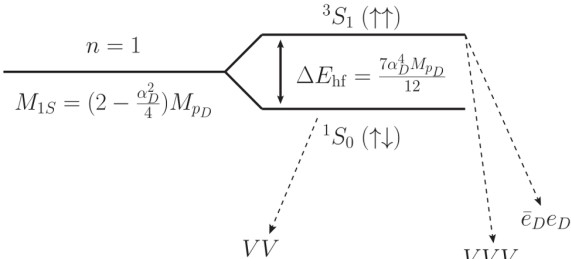

Figure 3: Diagramatic summary of the $\bar{p}_D p_D$ DM bound state decays, analogous to those of true muonium [59]. Bound state formation becomes important at relatively large values of $\alpha_D$ and $M_{p_D}$ and when the temperature drops far enough for the inverse ionization process to become slower than the relevant decay rates.

## 3  Relic Density

The required $\alpha_D$ to obtain the observed $\Omega_D$ from thermal freeze-out in the presence of an asymmetry is computed as in [51]. We take into account the Sommerfeld enhancement of the annihilation processes, as well as the formation and decay of $\bar{p}_D p_D$ bound states. New here is that the dark electrons provide an additional annihilation channel, shown in Fig. 2, and an additional decay channel for the spin-1 bound state. The annihilation and bound-state formation cross-sections are

$$v_{\text{rel}}\sigma_{\bar{p}_D p_D \to VV} = v_{\text{rel}}\sigma_{\bar{p}_D p_D \to \bar{e}_D e_D} = \frac{\pi\alpha_D^2}{M_{p_D}^2} \times S_{\text{ann}}, \tag{2}$$

$$v_{\text{rel}}\sigma_{\text{BSF}} = \frac{\pi\alpha_D^2}{M_{p_D}^2} \times S_{\text{BSF}}, \tag{3}$$

where $S_{\text{ann}}$ is the $s$-wave Sommerfeld enhancement factor, and $S_{\text{BSF}}$ arises from the appropriate convolution of the bound-state and scattering-state wavefunctions and thus includes the Sommerfeld effect. $S_{\text{ann}}$ and $S_{\text{BSF}}$ depend only on the ratio $\alpha_D/v_{\text{rel}}$ in the Coulomb approximation [60,61], which is satisfactory during freeze-out [44] (for a caveat, see [62] and Sec. 7.2). The decay widths of the spin-0 ($\uparrow\downarrow$) and spin-1 ($\uparrow\uparrow$) bound states are

$$\Gamma(\uparrow\downarrow \to VV) = \frac{\alpha_D^5 M_{p_D}}{2}, \tag{4}$$

$$\Gamma(\uparrow\uparrow \to \bar{e}_D e_D) = \frac{\alpha_D^5 M_{p_D}}{6}, \tag{5}$$

$$\Gamma(\uparrow\uparrow \to VVV) = \frac{2(\pi^2 - 9)\alpha_D^6 M_{p_D}}{9\pi}, \tag{6}$$

and are summarised in Fig. 3. Since $\Gamma(\uparrow\uparrow \to \bar{e}_D e_D) \gg \Gamma(\uparrow\uparrow \to 3V)$, the formation of $\bar{p}_D p_D$ bound states depletes the DM density more efficiently than in the absence of dark electrons. The $e_D, \bar{e}_D$ degrees of freedom also affect the dark-to-visible sector temperature ratio after the two sectors decouple. The combined effect of the dark electrons is to suppress the required $\alpha_D$. The DM mass is related to the proton mass $m_p$ by[4]

$$M_{p_D} = m_p \frac{Y_B}{Y_D} \frac{\Omega_{\text{DM}}}{\Omega_{\text{B}}} \left(\frac{1 - r_\infty}{1 + r_\infty}\right), \tag{7}$$

---

[4] For simplicity, here we assume that $m_{e_D} \ll M_{p_D}$, such that the dark electrons make up only a small fraction of the DM mass density. Of course, in the limiting case $m_{e_D} \approx M_{p_D}$, which may arise if the dark sector possesses some additional symmetry structure [58], the dark proton mass is smaller than shown in eq. (7).

where $Y_B$ is the baryon asymmetry, and $r_\infty \equiv (Y_-/Y_+)_{t\to\infty}$ is the ratio of DM antiparticles to particles today. Note that a smaller $r_\infty$ requires a larger annihilation cross-section [48, 51], thus larger $\alpha_D$.

Following Ref. [51], we employ unitarity – which sets an upper limit on the partial-wave inelastic cross-sections – to estimate the maximum mass for which DM can annihilate down to the observed density. In the present model, the direct annihilations are $s$-wave, while bound-state formation is $p$-wave. Using the corresponding leading order cross-sections (2) and (3), we find that the $s$-wave limit is saturated for lower $\alpha_D$. We employ this value to estimate the maximum DM mass, taking into account the depletion of DM via both the $s$-wave and $p$-wave inelastic processes. Since smaller $r_\infty$ requires more efficient annihilation, the maximum $M_{p_D}$ decreases with decreasing $r_\infty$ [51]. We note though that any computation close to the unitarity limit using (2) and (3) is likely inaccurate, since at those values of $\alpha_D$ higher order corrections should become significant, such that the cross sections remain below the unitarity bound for any value of $\alpha_D$ (thus causing the DM abundance to exceed its measured value).

## 4 Self Interactions

The cross section relevant for comparison to simulations of SIDM is the momentum transfer cross section, which should take into account both attractive, $p_D - \bar{p}_D$, and repulsive, $p_D - p_D$ or $\bar{p}_D - \bar{p}_D$, interactions. The attractive and repulsive cross sections are weighted with the appropriate densities, in order to obtain the effective transfer cross section,

$$\sigma_T = \frac{1}{2(n_\infty^{\text{sym}})^2}\left[n_\infty^+ n_\infty^- \sigma_{\text{att}} + \frac{1}{2}(n_\infty^+ n_\infty^+ + n_\infty^- n_\infty^-)\sigma_{\text{rep}}\right]$$
$$= \frac{2}{(1+r_\infty)^2}\left[r_\infty \sigma_{\text{att}} + \frac{1}{2}(1+r_\infty^2)\sigma_{\text{rep}}\right], \tag{8}$$

where $\sigma_{\text{att}}$ ($\sigma_{\text{rep}}$) is the attractive (repulsive) transfer cross section, $n_\infty^\pm$ is the DM (anti)-particle density today and $n_\infty^{\text{sym}}$ is the DM density today for symmetric DM of the same mass. We use the analytic approximations for $\sigma_{\text{att}}$ and $\sigma_{\text{rep}}$ given in Ref. [63]. The areas of sizable self interactions at velocities relevant for dwarf galaxies, $v_{\text{rel}} = 10^{-4}$, are shown as green bands in Figs. 1 and 4. For larger asymmetries (smaller $r_\infty$), the importance of the attractive interaction decreases, hence the parametric resonances begin to disappear. The bands also move up to slightly higher values of $M_V$ because $\alpha_D$ increases with decreasing $r_\infty$. Note $\sigma_T$ will be suppressed at velocities relevant for cluster collisions, $v_{\text{rel}} \approx 3 \times 10^{-3}$, due to the light mediator. Hence the constraints from systems with typically higher velocities [64] are naturally evaded.

In our computations, we neglect scatterings involving $e_D$ or $\bar{e}_D$. The effect of the dark electrons depends of course on their mass, and we do not attempt a detailed exploration along this parameter in the present work. We only briefly comment on the potential implications. In part of the parameter space, the dark electrons may increase the energy and momentum exchange inside halos, thereby shifting the preferred regions (green bands) towards higher $M_{p_D}$ and $M_V$ values, which are less constrained by indirect detection. In particular, since the dark electrons are lighter than the dark protons, the $p_D - e_D$ collisions are expected to be more frequent, albeit transferring less energy on average per event, than the $p_D - p_D$ scatterings. On the other hand, in large portions of the parameter space where scatterings involving dark electrons are expected to be significant, dark atoms may have formed in the early universe (cf. Sec. 7.1), thus screening the DM self-interactions today [54]. Very importantly, the effect of DM self-interactions in multicomponent DM models has not been simulated. While semi-analytical estimates are possible (see e.g. [54, 65, 66]), they involve larger-than-usual uncertainties. Our

work, in combination with the strong constraints on symmetric SIDM models [44, 45, 67], strongly motivate the development of multicomponent DM simulations.

Finally we note that for the range of mediator masses we consider here, $M_V > \text{MeV}$, dissipation is expected to be negligible. For studies of dissipation in various atomic DM scenarios, see e.g. [41, 42, 68].

## 5  Indirect Detection Constraints

Due to the suppressed population of DM antiparticles, the effective cross section for indirect detection is [48, 50, 51]

$$\sigma_{\text{ID}} \, v_{\text{rel}} \equiv \frac{n_\infty^+ n_\infty^-}{(n_\infty^{\text{sym}})^2} \sigma_{\text{inel}} \, v_{\text{rel}} = \frac{4r_\infty}{(1 + r_\infty)^2} \, \sigma_{\text{inel}} \, v_{\text{rel}} \,. \tag{9}$$

$\sigma_{\text{inel}}$ includes both the direct $\bar{p}_D p_D$ annihilation, and the formation of $\bar{p}_D p_D$ bound states, whose cross-sections are given by Eqs. (2) and (3). For the velocities relevant to the DM halos and the CMB, the factors $S_{\text{ann}}$ and $S_{\text{BSF}}$ depend strongly on the mediator mass, and have been computed in [69]. We use $\sigma_{\text{ID}}$ to set limits from PLANCK, AMS, FERMI, and ANTARES data, and present our results in fig. 4. Before discussing the various indirect probes further, we note that we determine the decay branching fraction of the mediator into SM final states, BR($V \to \bar{f}f$), according to the discussion in [44], which we follow closely here.

*Effect of the dark electrons.* In our analysis, we neglect the annihilations of the relic $\bar{e}_D e_D$ into dark photons. As long as $m_{e_D} \ll M_{p_D}$, the dark electrons annihilate more efficiently in the early universe, and have a smaller relic population available to annihilate at late times. In the symmetric limit, $r_\infty \simeq 1$, the number densities scale as $n_\infty(e_D)/n_\infty(p_D) \sim m_{e_D}/M_{p_D}$, while for $r_\infty \ll 1$, the residual symmetric population of dark electrons is more suppressed than that of the dark protons. Indeed, $r_\infty$ decreases exponentially with the annihilation cross-section [48, 51], thus $r_\infty(e_D) \ll r_\infty(p_D)$. A small $m_{e_D}$ may also imply that the fluxes from $\bar{e}_D e_D$ annihilations lie outside the energy ranges probed by FERMI, AMS and ANTARES. This last point does not apply to the PLANCK constraint. However, we find that the CMB constraint on $\alpha_D^2$ from $e_D$ annihilations is weaker by at least a factor $m_{e_D}/M_{p_D}$ than that from dark protons.[5]

It follows that the $\bar{e}_D e_D$ pairs produced in the $\bar{p}_D p_D$ annihilation and bound-state decay processes also do not contribute to our constraints. Their density is very small and the probability that they subsequently annihilate among themselves or with relic $\bar{e}_D$, $e_D$ is entirely negligible.

The above imply that the indirect constraints weaken due to the presence of the dark electrons in the theory, even for $r_\infty = 1$. This is because the dark electrons contribute to the depletion of DM in the early universe, thereby reducing the predicted $\alpha_D$, while they do not contribute to the indirect detection signals at late times.

*CMB constraints.* Dark matter annihilation can increase the ionization fraction of the plasma during the CMB epoch which can affect the measured anisotropies. In our model the effect is determined by the DM mass $M_{p_D}$, the relative velocity during the CMB, $v_{\text{rel}} \lesssim 10^{-8}$ [44],[6] the dark-photon branching fractions into SM final states, BR($V \to \bar{f}f$), which depend on $M_V$,

---

[5]This results from a combination of (i) the dependence of the relic number densities of $e_D$ and $p_D$ on their masses and of (ii) the fact that, for a given cross section dark electrons inject smaller energies than dark protons in the CMB plasma.

[6] The presence of the dark electrons, in the parameter space studied in this paper, cannot increase $v_{\text{rel}}$ sufficiently for $\sigma_{\text{inel}} v_{\text{rel}}$ to lie outside the saturated regime. Even though the dark electrons may mediate to delay the kinetic decoupling of the dark protons from the dark photons, the dark photons eventually also become non-relativistic, forcing the dark sector temperature to decrease as $(1 + z)^2$ thereafter. For $M_V \geq 1$ MeV, we find that $v_{\text{rel}} < 10^{-8}$ remains true around CMB. Hence, the thermally averaged cross section relevant for the CMB constraint is independent of the velocity, as in [44].

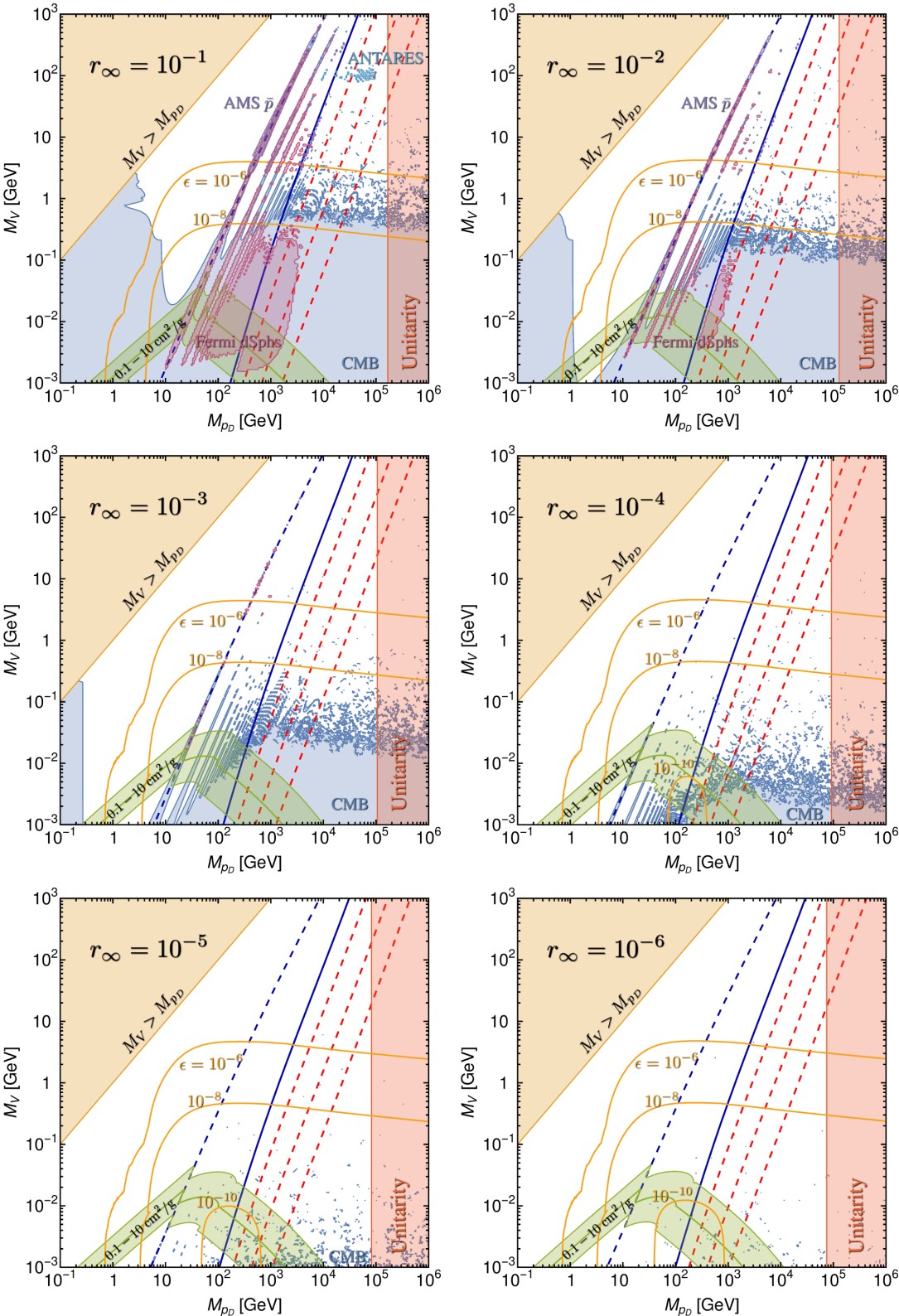

Figure 4: Same as for Fig. 1, but with decreasing values of the fractional antiparticle population, $r_\infty$. The indirect detection constraints still apply for moderate values of $r_\infty$, but the green shaded area (where sizable self-interactions are possible) eventually emerges.

and the final-state-dependent efficiency factor, $f_{\text{eff}}$ [70, 71]. We apply the constraint derived from the PLANCK data [7] to obtain the bounds shown in Figs. 1 and 4, where for $\sigma_{\text{ID}}$ we have used the Hulthen approximation regularised with the prescription of [72] to respect unitarity, see [44] for more details. For $M_{p_D} \gtrsim 100$ GeV and $r_{\infty}$-dependent values of $M_V$ (e.g. $M_V \approx$ GeV for $r_{\infty} = 1$) the region excluded by CMB is not precisely rendered in our figures because the resonances become increasingly dense. In these regions, the density of the coloured points provides a rough indication of the density of the actual resonances, and thus of the area excluded by the CMB. Constraints on ADM with contact only interactions from the CMB have been previously produced in [73]. They apply only to larger $r_{\infty}$ and a limited DM mass range.

AMS *antiprotons.* We use the method of [44]. Constraints arise for $M_V \gtrsim 2$ GeV for obvious kinematical reasons. Two parameter regions are constrained as clearly seen in Fig. 1. At $M_{p_D} \sim 10 - 100$ GeV, any additions to the well measured flux are tightly constrained, and at $M_{p_D} \gtrsim 100$ GeV, the resonances imply signals that would overwhelm the measured spectrum.

FERMI *Dwarfs.* We follow closely the analysis in [44], which is based on the FERMI observations of 15 dwarf galaxies, with gamma ray energies in the range 0.5 GeV to 0.5 TeV [74]. The statistical analysis treats the $J$–factors as nuisance parameters. Two separate regions are excluded (clearly seen in Fig. 1). In the Sommerfeld regime, the enhancement compensates in part for the suppression of the photon flux when $V$ decays mostly into leptonic channels.

FERMI *Galactic Halo.* Constraints from FERMI observations of the Galactic Halo were presented in [44], based on the analysis of [75]. We have updated the analysis considering a more recent dataset (Pass 8) and including more statistics. The constrained region disappears, except for some small areas close to the resonances. This is both due to the lower $\alpha_D$ predicted in the presence of the dark electrons, and because the bounds become slightly less severe with Pass 8 data. We have checked that allowing the overall background level to float does not change this conclusion. We do not show these small excluded regions in our figures.

ANTARES *neutrinos.* We derive constraints from ANTARES [76], which were not considered in [44], and are particularly stringent for $M_{p_D} \gtrsim 1$ TeV. The ANTARES bounds come from the non-observation, from the Milky Way halo, of excesses in muon neutrino fluxes. The existing constraints are cast for DM annihilating directly into SM pairs. Therefore we cannot directly use them, because in our model DM annihilates first into $V$ that then decay into SM particles, resulting in a one-step cascade that softens and broadens the spectra. We overcome this limitation as follows.

First, we compute the final $\nu_\mu$ spectra from DM annihilations directly into all SM pairs constrained by ANTARES, using the PPPC4DMID [77] with the electroweak corrections switched on [78] (which was used also by the ANTARES collaboration in [76]). We also compute neutrino spectra from one-step cascades in the simple limit $M_V \ll M_{p_D}$ (see e.g. [79] for more details), which holds in the region where the bounds will apply. Then, we compare the spectra obtained with the above two procedures, and notice that the high energy $\nu_\mu$ fluxes – which drive the ANTARES limit[7] – are the least changed for quark-antiquark final states, and that the cascade yields a similar amount of high-energy neutrinos. Therefore, we exclude regions of our model in which

$$\text{BR}(V_D \to \bar{f} f) \langle \sigma_{\text{ID}} v_{\text{rel}} \rangle > \langle \sigma_{\text{DM DM} \to \bar{f} f} \, v_{\text{rel}} \rangle^{\text{ANTARES}}_{\max}, \tag{10}$$

for at least one of the two final states $\bar{f} f = \bar{b} b$ plus lighter quarks, and $\bar{f} f = \bar{\nu}_{e,\mu,\tau} \nu_{e,\mu,\tau}$. For the $\bar{b} b$ channel, we sum over all lighter quarks, as the spectra are very similar. Following ANTARES we also take into account the neutrino oscillations.

---

[7] We infer this from the fact that the limits of Ref. [76]: (i) strengthen at larger DM masses; (ii) are almost identical for $\mu^+ \mu^-$ and $\tau^+ \tau^-$ final states, whose resulting $\nu_\mu$ spectra are almost identical at high-energy but differ substantially at low energies; (iii) are much stronger on lepton than on $b\bar{b}$ final states, and the latter are peaked at lower energies than the former.

Our constraints are conservative in the sense that, in our model, one would have to sum over all final states $\bar{f}f$, instead of constraining one final state at a given time. This procedure is impracticable here because it would require to perform the ANTARES analysis using, as signal flux, the specific one coming from our model, which is not immediately comparable with the flux from the final states considered by ANTARES. On the other hand, our constraints would be relaxed by the choice of a more cored profile than the one used in [76]. While [76] shows the impact of different profile choices for the $\tau^+\tau^-$ final state, it does not do the same for the other ones. Therefore we choose to consistently use the only profile for which the limits is expressed for all final states.

The excluded area is shown in Figs. 1 and 4. It disappears somewhere between $r_\infty = 10^{-1}$ and $r_\infty = 10^{-2}$. Following closely the procedure of Ref. [44] for the dark photon decays, we find another excluded region for 200 MeV $\lesssim M_V \lesssim$ 5 GeV and $M_{p_D} \gtrsim$ 3 TeV. However, we believe that this exclusion is, at least partly, an artifact of the way dark photon decays are modeled, in the $M_V$ region where hadron final states are important (i.e. exactly in the $M_V$ region just mentioned). Indeed, in Ref. [44], the dark photon was assumed to decay with 50% probability into muons, and 50% in neutrinos, to reflect the dominant final states from $\pi^\pm$ etc. The spectra of $\mu$ and $\nu_\mu$ would, in reality, be softer, because of the additional steps (the $\pi^\pm$) involved. Therefore, consistently with our understanding that the limits are driven by the high-energy part of the spectra, we conservatively choose to not show on our plots the ANTARES exclusion in that region.

*Other limits from high energy cosmic-rays.* Our previous inclusion of ANTARES limits demonstrates that high energy cosmic-ray experiments probe otherwise unexplored regions of parameter space of this model. In this domain of energies, we have not attempted to recast recent limits from ICECUBE [80], VERITAS [81] and HAWC [82, 83] to secluded models. This is motivated by the following: i) it is difficult, based on [80–83], to derive which energies of the measured cosmic-rays dominate the exclusions (see footnote 7 for ANTARES), which was a necessary ingredient for our simple recast of the ANTARES limits; ii) the limits on standard DM annihilations, given in [80–83], are much weaker than those given by ANTARES.

This second argument however does not hold for recent HESS limits [84] from 254 hours of observation of the GC. Nonetheless, to recast them for secluded DM models we could not use the same prescription we used for ANTARES limits, because of reason i) above. To recast HESS limits, a possibility would be to directly analyze HESS data. To our knowledge, however, these data are not public. To overcome this lack of public data, Ref. [85] used public data of the 2011 HESS limits on DM annihilations [86], from 112 hours of observation of the GC, analysed them for secluded models, and projected them to 254 hours. The HESS reach obtained in this way is potentially very constraining. We do not include it in our study, however, because it is partly a projection. Moreover, unlike any other bounds we included, HESS loses any sensitivity if the DM density distribution features a core at the GC as small as 300 pc, which is currently allowed from the observational point of view (see e.g. [87, 88]) and not tested by current numerical simulations (see e.g. [89–92]). Finally, the HESS reach from [85] does not probe qualitatively new mass regions, the highest DM mass probed being 10 TeV.

We do encourage the HESS collaboration, as well as other high energy cosmic-ray telescopes, to cast their limits also on secluded DM models, and to present their results in a format that could easily be reinterpreted in different DM scenarios.

## 6  Direct Detection

We compute the constraints from CRESST-II [8], CDMS-LITE [9], and LUX [11], each of which in turn gives the strongest bound for increasing $M_{p_D}$. The constraints for different values of $\epsilon$ are

shown as yellow contours in Figs. 1 and 4. As $r_\infty$ is decreased, the required $\alpha_D$ is increased, and the direct detection constraints become stronger. Hence, in Fig. 4 the $\epsilon = 10^{-10}$ contour appears for $r_\infty = 10^{-4}$. The method used in setting the constraints from CRESST-II and LUX is given in Appendix B of [93] (also see [52, 53, 94]).[8] We use a similar method in deriving the constraint for CDMS-LITE, the details of which are given in the appendix. We checked that the constraints from XENON1T [12] and PANDAX-II [13] are similar to those arising from LUX.

## 7 Caveats

### 7.1 Dark atomic bound states

Due to the presence of the dark electrons, DM may form Hydrogen-like bound states. The radiative capture of $p_D$ and $e_D$ into a dark atom with emission of a mediator $V$ is possible if the binding energy is larger than the mediator mass,

$$\frac{\alpha_D^2}{2} \frac{M_{p_D} m_{e_D}}{(M_{p_D} + m_{e_D})} > M_V. \tag{11}$$

If dark recombination is efficient in the early universe, then the amount of dark protons available to annihilate today may be significantly reduced, thereby suppressing the expected annihilation rate, as well as the direct detection signals. A proper treatment then requires computing the residual ionization fraction of DM today. This is beyond the scope of the present work. (The cosmology of atomic DM with a massless dark photon has been computed in detail in Ref. [65].) Instead, in Figs. 1 and 4, we simply mark the regions where dark atomic bound states may form. Below the red dashed contours, for the indicated values of $m_{e_D}$, our direct and indirect constraints are ameliorated, while the DM self-interactions are suppressed.

### 7.2 Resonances and Reannihilation

It has recently been pointed out that, on the parametric resonances of the Sommerfeld enhancement factor, a two stage freeze-out may occur [62]. The first stage is the standard freeze-out and happens in the Coulomb limit of the Yukawa potential, at $M_{p_D}/T \sim 20$ or $v_{\rm rel} \sim 0.1$, in which the parametric resonance is not yet manifest. On the parametric resonances, the cross section scales as $\sim 1/v_{\rm rel}^2$ for small velocities, before saturating. The DM annihilation then comes back into equilibrium, after the DM has kinetically decoupled, resulting in a second period of significant DM density depletion [95–98]. This second stage of the freeze-out process is known as reannihilation.

As our freeze-out calculation does not take into account the period of reannihilation, the values of $\alpha_D$ found on resonance are slightly too large and would result in a DM underabundance. The correct procedure would be to detune $\alpha_D$ on these points, obtain an overabundance in the first stage of freeze-out, and then reach the correct relic abundance through reannihilation. Our derived constraints on resonance are therefore somewhat too strong. However, moving significantly off resonance means suppressing the reannihilation effect, as the cross section then grows only as $\sim 1/v_{\rm rel}$ at small velocities. The detuning away from the resonance peak is therefore necessarily small. Further detailed numerical work is required to determine both the amount of detuning and the quantitative impact of reannihilations on the annihilation cross section relevant for indirect detection.

---

[8] Conservatively, for CRESST-II, we do not apply the convolution of the recoil energy with the 62 eV width Gaussian, as described in [93]. This results in a weaker constraint for low $M_{p_D}$, because sub-threshold events from the steeply falling spectrum are not shifted into the detectable range. Our overall conclusions are not affected by this choice.

Off resonance, our indirect detection constraints are more robust. As it was shown in [62], however, by choosing $\alpha_D$ progressively closer to the peak on lower resonances, it can still be possible to obtain the correct relic abundance from multiple values of $\alpha_D$ for the same $(M_{p_D}, M_V)$ parameter point. Hence, also in this case, further work must be done in order to derive the constraints at these lower $\alpha_D$ values. However, note that the indirect detection constraints are the strongest on resonance, so the smaller $\alpha_D$ values may not open up considerable areas of parameter space. The quantitative determination of these effects, which were also ignored in [44, 45], is left for future work.

### 7.3 Further Constraints on the Mediator

The results here have been presented in the $M_{p_D} - M_V$ plane. To confirm that SIDM is possible in this model, we must also compare the allowed areas to those on the $M_V - \epsilon$ plane in Fig. 5 of [44]. For mediator masses relevant for SIDM, between $10^{-3} \lesssim M_V/\text{GeV} \lesssim 10^{-1}$, there are allowed areas centred around $\epsilon \sim 10^{-10}$ and spanning one or two orders of magnitude. Lower (higher) values of $\epsilon$ are ruled out by BBN (Supernova 1987A [99]).[9] From Fig. 4, this range of $M_V$ is also allowed by direct detection experiments and ID, provided the asymmetry is large enough. Hence SIDM is indeed viable in this model.

### 7.4 On the assumption of early thermal equilibrium

All our results have been derived under the assumption of early thermal equilibrium between the dark sector and the SM, $T_D = T_{\text{SM}}$, for temperatures larger than any of the other scales we have considered. This is justified for values of the kinetic mixing $\epsilon \gtrsim 10^{-6}$, see e.g. [44, 100], although this condition may weaken due to the presence of dark electrons. The values of $\epsilon$ allowed by direct detection, therefore, may not be large enough to realise early equilibrium in the region of parameter space relevant for self-interactions (see Figs. 1 and 4). In that region, the assumption of early thermal equilibrium could be justified by the presence of some extra dynamics that connects the SM and the dark sector. These dynamics could lie at an energy scale large enough to not alter the phenomenological study we have carried out, and would be even welcome to give a common origin to the baryon and Dark Matter asymmetries.

In the absence of such dynamics, an early $T_D/T_{\text{SM}} \neq 1$ would result in a freeze-out abundance different with respect to the one we have computed here. This would in turn modify the values of $\alpha_D$ in our parameter space, and thus the regions corresponding to sizeable self-interactions and to exclusions from indirect and direct detection. A more detailed study of the impact of different thermal histories go beyond the purposes of this paper.

## 8 Conclusions

The focus of this paper has been to derive constraints on a relatively simple, well-motivated example of a dark sector, featuring fermionic DM and a light massive mediator, and allowing for the possibility of a dark asymmetry.

We have demonstrated that ADM coupled to light force mediators provides a viable framework for SIDM. While the combined direct, indirect and cosmological bounds severely limit the possibility of symmetric thermal-relic SIDM [44, 45, 67], large portions of the ADM parameter space evade the indirect constraints, due to the suppressed annihilation rate.

Nevertheless, we have shown that ADM with a Sommerfeld enhancement can yield significant annihilation signals that are constrained by various astroparticle messengers, i.e. $\gamma$-rays,

---

[9]For few eV $\lesssim M_V \lesssim$ MeV the BBN bound becomes much more stringent.

antiprotons and neutrinos. These signals remain important even for large asymmetries, in particular late-time antiparticle-to-particle ratios as low as $r_\infty \sim 10^{-4}$, in our model. This challenges the standard lore that ADM cannot be probed via ID. Level transitions may give rise to additional radiative signals in this class of models [101–104]. Importantly, the ID constraints derived here extend beyond the SIDM regime. We encourage high energy cosmic-ray experiments to present their results in a way amiable for reinterpretation in secluded DM models, e.g. in terms of flux as a function of energy. Future improvements to direct detection experiments will be important in testing this scenario more thoroughly.

Asymmetric DM models that feature light or massless force carriers result typically in multi-component DM, due to various considerations, including our arguments from gauge invariance and due to possible formation of stable bound states. Since ADM is a natural framework for SIDM, there is strong incentive for the development of multicomponent DM simulations.

## Acknowledgements

I.B. thanks Kai Schmidt-Hoberg and Sebastian Wild for helpful discussion. M.C. acknowledges the hospitality of the Institut d'Astrophysique de Paris (IAP) where a part of this work was done.

**Funding information** This work has been done in part within the French Labex ILP (reference ANR-10-LABX-63) part of the Idex SUPER, and received financial state aid managed by the Agence Nationale de la Recherche, as part of the programme *Investissements d'avenir* under the reference ANR-11- IDEX-0004-02. The work of M.C. is supported by the European Research Council (ERC) under the EU Seventh Framework Programme (FP7/2007-2013) / ERC Starting Grant (agreement n. 278234 - 'NEWDARK' project). The work of M.T. is supported by the Spanish Research Agency (Agencia Estatal de Investigación) through the grant IFT Centro de Excelencia Severo Ochoa SEV-2016-0597 and the projects FPA2015-65929-P and Consolider MultiDark CSD2009-00064. K.P. was supported by the ANR ACHN 2015 grant ("TheIntricateDark" project), and by the NWO Vidi grant "Self-interacting asymmetric dark matter".

## A CDMS-lite

In setting the limit from CDMS-LITE [9], we use a similar technique as for CRESST-II. We use Poisson statistics to calculate the 90% CL on the number of nuclear recoils, assuming the events observed by CDMS-LITE between $E_{\min}$ and $E_{\max}$ are due to DM induced nuclear recoils. Here $E_{\min} = 56 \, \text{eV}_{\text{ee}}$ is the threshold energy of the detector and

$$E_{\max} = \text{Min}\left[ \frac{2m_{\text{DM}-T}^2}{m_T}(v_{\text{E}} + v_{\text{Esc}})^2, \, 1 \, \text{keV}_{\text{ee}} \right], \tag{12}$$

where $v_E$ ($v_{\text{Esc}}$) is the velocity of the earth (escape velocity) with respect to the DM halo, $m_T$ is the target mass, and $m_{\text{DM}-T}$ is the reduced mass of the DM and target. That is, $E_{\max}$ is the either the maximum possible recoil energy given the DM mass or $1 \, \text{keV}_{\text{ee}}$ — depending on which is smaller. The $1 \, \text{keV}_{\text{ee}}$ cutoff is chosen because above this the large number of counts due to the $1.3 \, \text{keV}_{\text{ee}}$ $L$-shell $^{71}$Ge electron-capture decay line will begin to contribute as background. Note the CDMS-LITE events are reported in electron equivalent recoil energies. We convert the values into nuclear recoil energies, $\text{keV}_{\text{nr}}$, using the central value of the Lindhardt model as used by CDMS-LITE ($k = 0.157$). We then demand the expected number of events our model

will contribute at a given point in parameter space, also taking into account the efficiency of CDMS-LITE, not exceed the 90% CL on the number of signal events we have calculated.

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
