# Peer review of "Asymmetric dark matter: residual annihilations and self-interactions"

_SciPost Physics, doi:SciPost Phys. 4, 041 (2018)_

## Round 2 · Referee Report · Anonymous · 2018-4-17

Strengths
1- Interesting model, novel results;
2- Very clear presentation;
3- Many connections to other works and open research problems.
Weaknesses
1- Incomplete discussion of the effects of dark electrons;
2- Room for improvements in the calculation and presentation of Sommerfeld-enhanced CMB constraints;
3- Missing discussion of non-standard temperature ratios.
Report
In their work the authors study a specific model of DM, which has three essential ingredients: a dark photon mediator from a new broken gauge group, a particle-antiparticle asymmetry in the dark sector and two oppositely charged DM species, named dark protons and dark electrons. The study provides an interesting new perspective on atomic dark matter by considering the case where the mediator is sufficiently massive that recombination in the dark sector is either impossible or inefficient, so that (repulsive) self-interactions can be very important at late times. At the same time the mediator is sufficiently light that these self-interactions are enhanced by non-perturbative effects.
Similar models without an asymmetry turn out to be strongly constrained by indirect detection and CMB constraints and the purpose of the present study is to investigate how strongly the symmetric component must be suppressed in order to evade these constraints. It turns out that a rather strong suppression is necessary, implying that even models with a strong particle-antiparticle asymmetry may be testable with these observations.
The manuscript is very well written and the presentation is clear. The results are robust and of interest to the community. In short, I can recommend publication once the following issues have been addressed.
Requested changes
1- It is well known that for the dark photon model direct detection constraints forbid thermal equilibrium between the dark sector and the Standard Model sector in the interesting regions of parameter space. The authors should comment, at least qualitatively, on the effect of a temperature ratio different from unity. Such a discussion would be a valuable addition to section 7.
\[\]
2- Throughout the work the authors assume that the dark electrons play no role apart from ensuring charge neutrality. It seems to me that there are a number of important caveats to this assumption. First, I would expect the dark electrons to have stronger self-interactions than dark protons (at least in regions of parameter space where the mediator is effectively massless). This could lead to observable consequences unless the dark electrons are at least an order of magnitude lighter than the dark protons, such that their contribution to the total dark matter abundance is negligible. If dark electrons are assumed to be at least an order of magnitude lighter than dark protons, this implies that also dark photons need to be this light, to ensure efficient annihilation of the symmetric component of dark electrons. This means that the inaccessible region at the top left of the plot should be much larger than currently indicated. Another potential concern is that the dark proton freeze-out will lead to an increase of the dark sector temperature relative to the temperature of the Standard Model sector. Freeze-out of dark electrons may then be less efficient, leading to a larger abundance of the symmetric component. Again, this issue presumably disappears for sufficiently light dark electrons, at the expense of requiring sufficiently light dark photons. The corresponding discussion needs to be extended.
\[\]
3- I wonder where the statement $v_\text{CMB} \sim 10^{-8}$ stems from. Presumably, this requires a calculation of the kinetic decoupling temperature, which should depend on additional details of the model, such as the mass of the dark electrons. I would suggest that the authors clarify the origin of this value and to what degree their results depend on the kinetic decoupling temperature (presumably not very much).
\[\]
4- In the parameter region where the resonances from Sommerfeld enhancement become very densely spaced, it becomes difficult to indicate excluded parameter regions, leading to CMB constraints that look quite ugly. I am wondering whether there may be a more meaningful way of drawing constraints, such as indicating in some way the density of resonances (i.e. how likely it is that a randomly chosen nearby point will be excluded because it is sufficiently close to a resonance). I realize that this may require substantial additional work, so maybe it is sufficient to add a comment on how the reader should interpret the CMB constraints in these regions.
\[\]
5- In the calculation of these CMB constraints, have the authors included the modification of the Sommerfeld enhancement factor due to unitarity restoration (see arXiv:1603.01383)? Or is this effect not relevant here?
\[\]
6- The authors state that they find a second excluded region from ANTARES with $M_{p_D} \geq 3 \times 10^3 \, \mathrm{TeV}$. This is presumably a typo -- should this read GeV?
\[\]
7- The authors encourage indirect detection experiments to perform an interpretation of their data in the context of this model. I personally think it would be even more important to encourage them to make data available in a format that allows for an independent interpretation. Maybe the authors would like to consider modifying this statement?

---

## Round 2 · Referee Report · Anonymous · 2018-4-30

Strengths
1 interesting model with rich phenomenology
2 very clear presentation
Weaknesses
mentioned below
Report
The authors study an interesting two-component asymmetric self-interacting dark matter scenario coupled to a light U(1) vector boson.
A particular focus of the current study is its astrophysical and cosmological implications to infer the viable regions of parameter space.
Dark matter self-interactions received a lot of interest lately, in particular models which allow for a velocity dependence
of the scattering cross section. The simplest realisations however are in strong tension with CMB
limits (vector mediators) or a combination of BBN and direct detection limits (scalar mediators).
The current paper is an interesting attempt to avoid these constraints, in particular the
constraints from late time energy injection due to dark matter annihilations by assuming an asymmetry in the dark sector which suppresses the annihilation rate.
The authors provide a very comprehensive analysis and the paper is very well written. It should be considered for publication in
SciPost after a couple of minor comments have been addressed.
Requested changes
It is not completely clear what the role of the dark electron is. If the dark electron mass is close to the dark proton mass it should
contribute sizably to the dark matter abundance. Equation (7) seems to indicate that the DM abundance is dominated by the dark protons alone.
Does this mean that the dark electrons are much lighter? If so does the electron abundance need to be reduced via annihilations (in which case
the dark photon would need to be even lighter) or is it OK (e.g. with $N_\text{eff}$) if a larger symmetric component remains?
If the dark electron is very light does it induce sizable dissipation? It would be good to discuss these points a bit more.
The kinetic mixing parameter is strongly constrained by direct detection and does not allow for thermalisation of the two sectors in large regions of the parameter space.
Without further assumptions the temperatures of the two sectors are therefore independent. The authors should comment on the implications of this.

---

## Round 3 · Referee Report · Anonymous (Referee 2) · 2018-5-28

Report

The authors have largely addressed the questions and comments of the referees and the manuscript is now suitable for publication.

---

## Round 3 · Referee Report · Anonymous (Referee 1) · 2018-5-29

Report

The authors have made a number of small changes to address my previous comments. I agree that it makes sense not to extend the discussion further in order to keep the paper focused. While I am happy to recommend the paper for publication, I do have one final comment.

The authors write in the new submission that "the density of the coloured points provides a rough indication of the density of the actual resonances". I'm wondering whether this statement is based on an actual understanding of how the plotting algorithm works, or if this is simply based on the assumption that plot points are drawn randomly. Should the authors plan a follow-up study, I would encourage them to think about this issue more carefully.

---

## Round 3 · Author Response

We thank the anonymous referees for their positive feedback on our paper and for their comments and suggestions. We address them below, explaining the related changes we implement in the revision version of our paper that we are submitting. The comments of the referees have allowed to us to improve the quality of our paper, and we hope the revised version can now be accepted for publication.

---

## Round 3 · List of Changes

Reply to Report 1

  1. The referee is right that direct detection (DD) constraints forbid early thermal equilibrium between the dark sector and the SM, in the region of parameter space interesting for Self-Interacting DM. We would like to observe that instead, for $m_V >$ few GeV, DD allows for $\epsilon \gtrsim 10^{-6}$, and therefore for early thermal equilibrium between the SM and the Dark sector (see e.g. footnote 8 of 1612.07295). That region is not interesting for SIDM, but it is interesting to study the impact of an asymmetry on some ID probes of this model, which is one of the purposes of our paper. We would also like to observe that the above considerations assume that only the kinetic mixing is responsible for the communication between the two sectors. It is possible that extra dynamics kept the two sectors in thermal equilibrium at large enough temperatures. Such dynamics could arise at an energy scale $\gg M_{p_D}, m_{\rm SM}$ and have no effect on the phenomenology we studied, and it would be somehow expected if one would want to link the origin of the baryon and dark matter asymmetries.

Following the referee's suggestion, we have added in Section 7 a synthesis of the above considerations, and a qualitative discussion of the effects, on our findings, of non thermal equilibrium.

  1. We agree with the referee that the dark electrons may have various implications. Although we comment in our manuscript on some significant aspects, a detailed exploration of the effects of the dark electrons is beyond the scope of this work. With respect to the points raised by the referee:

2a. DM self-interactions due to the dark electrons. Please see last paragraph of section 4. We have added a clarifying remark.

2b. Maximum dark photon mass. We agree that for the efficient annihilation of the dark antielectrons, the dark photons have to be lighter than the dark electrons, and the inaccessible region of our parameter space grows. We have added a remark in the caption of fig. 1. We note though that even if $m_{e_D}$ is smaller than $M_{p_D}$ by several orders of magnitude, the parameter space for self-interacting DM remains largely open or even unaffected.

2c. Dark electron freeze-out. In our freeze-out computation, we have assumed that the thermal equilibrium between the SM and the dark sector ceases at temperatures larger than the dark proton mass (see comment above), and therefore we have taken into account the reheating of the dark sector due to $p_D$ becoming non-relativistic. This occurs before even the dark protons freeze-out, and affects the efficiency of their annihilation as much as that of the dark electrons.

The $e_D$ freeze-out may be further affected by the dark sector reheating due to $e_D$ becoming non-relativistic. This is a very mild effect, since the required cross-section scales as $\sigma_{\rm ann} v_{\rm rel} \propto T_D/T_{\rm SM} \propto g_{*,D}^{-1/3}$. This effect is far outweighed by the fact that the $e_D$ annihilation cross-section is larger than that of $p_D$, due to the smaller mass of the former. Quite generally, $r_\infty (e_D) \leqslant r_\infty (p_D)$.

  1. We intended to write $v_{\rm rel} \lesssim 10^{-8}$ at the CMB epoch, we apologize for the typo, that we corrected. This value comes from the more detailed analysis in 1612.07295 (our ref [44], see Sections 3.1 and 4.3 therein). Dark electrons do not change the outcome of that analysis and we have added a footnote about this for completeness. Although, in presence of dark electrons the kinetic decoupling of the dark protons from the dark photons can be delayed, we have estimated that $v_{\rm rel}$ remains $< 10^{-8}$ at the CMB epoch.

  2. We have added in Section 5 a comment as suggested by the referee, on the visualization of the CMB exclusion in the region where resonances are most dense.

  3. Yes, in the calculation of the CMB constraints we have included the modification of the Sommerfeld enhancement factor due to unitarity restoration. We have added an explicit comment about this, again in Section 5.

  4. The referee is right, that is a typo, we have now removed the extra $10^3$.

  5. We agree with the referee, and we have added the encouragement suggested to the relevant sentence in Section 5. We also note that the general encouragement suggested by the referee, to present data in a way that it is amiable for reinterpretation, is also present in the Conclusions.

  6. Finally, we have updated the plots after correcting an efficiency factor for the CDMSlite analysis, which resulted in slight changes to the direct detection constraints.

Reply to Report 2

Please see the response to Report 1 (item 2) for comments related to the dark electrons. With respect to the additional points raised:

If $m_{e_D} \approx M_{p_D}$, then the dark electrons should be included in the computation of the DM relic density and the estimation of the dark proton mass. This is a limiting case, which may arise due to a particular symmetry structure in the dark sector, and we shall not devote a specific analysis here. We have added a footnote about this in section 3.

The dissipation is a rather complex issue that depends sensitively of the energy splittings, interactions rates and the thermodynamic conditions. Even for massless mediators, it happens efficiently only in a very limited part of the parameter space. In our analysis, we focus on fairly massive mediators, $m_V > {\rm MeV}$, for which dissipation is expected to be negligible. We have added a comment in the end of section 4, and provided references to related studies.

The thermalisation of the two sectors is addressed by point 1 in the reply to the first referee report.

---

## Editorial Decision

published